# Multi-Drug Cocktail Therapy Improves Survival and Neurological Function after Asphyxial Cardiac Arrest in Rodents

**DOI:** 10.3390/cells12111548

**Published:** 2023-06-05

**Authors:** Rishabh C. Choudhary, Muhammad Shoaib, Kei Hayashida, Tai Yin, Santiago J. Miyara, Cristina d’Abramo, William G. Heuser, Koichiro Shinozaki, Nancy Kim, Ryosuke Takegawa, Mitsuaki Nishikimi, Timmy Li, Casey Owens, Ernesto P. Molmenti, Mingzhu He, Sonya Vanpatten, Yousef Al-Abed, Junhwan Kim, Lance B. Becker

**Affiliations:** 1Laboratory for Critical Care Physiology, Feinstein Institutes for Medical Research, Northwell Health, Manhasset, NY 11030, USA; rchoudhary1@northwell.edu (R.C.C.);; 2Institute of Bioelectronic Medicine, Feinstein Institutes for Medical Research, Manhasset, NY 11030, USA; 3Department of Emergency Medicine, Northwell Health, Manhasset, NY 11030, USA; 4Donald and Barbara Zucker School of Medicine at Hofstra/Northwell, Hempstead, NY 11549, USA; 5Elmezzi Graduate School of Molecular Medicine, Manhasset, NY 11030, USA; 6Litwin-Zucker Center for Research in Alzheimer’s Disease, Feinstein Institutes for Medical Research, Northwell Health, Manhasset, NY 11030, USA; 7Department of Emergency Medicine, Donald and Barbara Zucker School of Medicine at Hofstra/Northwell, Hempstead, NY 11549, USA; 8Department of Surgery, Northwell Health, Manhasset, NY 11030, USA; 9Department of Molecular Medicine, Donald and Barbara Zucker School of Medicine at Hofstra/Northwell, Hempstead, NY 11549, USA; 10Emergency Medicine, Feinstein Institutes for Medical Research, Northwell Health, 350 Community Dr., Manhasset, NY 11030, USA

**Keywords:** brain injury, cell death, cardiac arrest, cardiopulmonary resuscitation, cardiopulmonary bypass resuscitation, ischemic damage, multi-drug cocktail, neurodegeneration, neuroprotection, return of spontaneous circulation

## Abstract

Background: Cardiac arrest (CA) can lead to neuronal degeneration and death through various pathways, including oxidative, inflammatory, and metabolic stress. However, current neuroprotective drug therapies will typically target only one of these pathways, and most single drug attempts to correct the multiple dysregulated metabolic pathways elicited following cardiac arrest have failed to demonstrate clear benefit. Many scientists have opined on the need for novel, multidimensional approaches to the multiple metabolic disturbances after cardiac arrest. In the current study, we have developed a therapeutic cocktail that includes ten drugs capable of targeting multiple pathways of ischemia–reperfusion injury after CA. We then evaluated its effectiveness in improving neurologically favorable survival through a randomized, blind, and placebo-controlled study in rats subjected to 12 min of asphyxial CA, a severe injury model. Results: 14 rats were given the cocktail and 14 received the vehicle after resuscitation. At 72 h post-resuscitation, the survival rate was 78.6% among cocktail-treated rats, which was significantly higher than the 28.6% survival rate among vehicle-treated rats (log-rank test; *p* = 0.006). Moreover, in cocktail-treated rats, neurological deficit scores were also improved. These survival and neurological function data suggest that our multi-drug cocktail may be a potential post-CA therapy that deserves clinical translation. Conclusions: Our findings demonstrate that, with its ability to target multiple damaging pathways, a multi-drug therapeutic cocktail offers promise both as a conceptual advance and as a specific multi-drug formulation capable of combatting neuronal degeneration and death following cardiac arrest. Clinical implementation of this therapy may improve neurologically favorable survival rates and neurological deficits in patients suffering from cardiac arrest.

## 1. Introduction

Each year, more than 356,000 individuals in the United States die due to cardiac arrest (CA), and nearly 90% of them suffer from poor neurological outcomes, even upon temporarily successful resuscitation [1,2]. Despite the alarming levels of mortality, there has been insufficient progress made in the development of pharmacological interventions that can significantly enhance post-CA survival and neurological outcomes. CA is a systemic ailment in which the cessation of blood flow to tissues throughout the whole body results in global ischemia [3]. Further, the sudden reintroduction of oxygen during resuscitation causes reperfusion injury, which exacerbates the prior ischemic damage. Upon ischemia–reperfusion injury, a complex cascade of metabolic dysfunction is observed [4], including endoplasmic reticulum stress [5], mitochondrial dysfunction [6], oxidative injury [7], depletion of high-energy metabolites such as ATP [8], and inflammation [9]. These events may cause brain injury and cell death, leading to impaired neurological function and poor survival outcomes. Furthermore, metabolomic profiling of rat tissue as well as rat and human plasma after prolonged CA revealed resuscitation-mediated metabolic dysfunction that indicates multi-organ injury [10,11]. Thus, the multifaceted pathogenesis of post-CA ischemia–reperfusion injury may necessitate a combination therapy approach for more successful treatment.

Although numerous studies have investigated drugs that potentially offer neuroprotective benefits after ischemic reperfusion injury [12], many of them employ a single-drug approach. Although this approach is useful in comprehending the disease’s pathology and generating data for developing therapies for diseases that can be treated with single-drug therapies, it seems that using only a single drug is insufficient for offering protection against a prevalent disease such as CA. As a disease with a global systemic impact, CA involves a multitude of metabolic, energetic, and inflammatory pathway alterations. This complexity and severity must be addressed with a more comprehensive, multi-drug approach. Multi-drug cocktails have a history of use for the treatment of other complex diseases, such as HIV/AIDS [13] and cancer [14]. In spite of this, multi-drug approaches have not yet been thoroughly explored in CA management. Currently, a few studies have reported that combined drugs for a single pharmacologic intervention or with therapeutic hypothermia as a post-asphyxial CA therapy [15,16,17]. However, these studies have two major limitations: (1) in using the fewest possible drugs, the majority of the injury pathways are unable to be targeted, and (2) these drugs have not undergone testing in severe asphyxial CA models, which necessitates a more comprehensive therapeutic approach.

We conducted a blind, randomized, placebo-controlled study with the objective of creating a multi-drug cocktail and evaluating its effectiveness in improving survival and neurological outcomes at 72 h post-resuscitation in a severe asphyxial CA rodent model.

## 2. Methods

### 2.1. Experimental Design

#### 2.1.1. Scientific Rationale for the Use of a Multi-Drug Cocktail

After a comprehensive search of the literature, we selected a total of 11 potential neuroprotective drugs for the formulation of the multi-drug cocktail, which can target multiple pathways simultaneously. We broadly categorized the drugs into four different groups: antioxidants (coenzyme Q10 (CoQ10) (Cayman Chemical, Ann Arbor, MI, USA) Edaravone (Sigma-Aldrich, St. Louis, MO, USA), N-acetyl cysteine (Sigma-Aldrich, St. Louis, MO, USA), and vitamin C (Cayman Chemical, Ann Arbor, MI, USA)); calcium and ionic controls (sodium amobarbital (Lipomed, Cambridge, MA, USA) and Zoniporide (Toronto Research Chemicals, Toronto, ON, Canada)); lipid membrane regulation (poloxamer 188 ((Sigma-Aldrich, St. Louis, MO, USA), and Szeto–Schiller peptide-31 (SS-31)); and mitochondrial protectant and energy production (ATP-magnesium chloride (MgCl_2_) (Sigma-Aldrich, St. Louis, MO, USA), cyclosporine A (Cayman Chemical, Ann Arbor, MI, USA), metformin (Sigma-Aldrich, St. Louis, MO, USA), and sulbutiamine (Toronto Research Chemicals, Toronto, ON, Canada)). For developing a multi-drug cocktail, each drug had a strong rationale for its use based on previously published reports (Table 1).

As such, CoQ10, also known as ubiquinone, is considered a strong antioxidant [18]. The neuroprotective effect of CoQ10 has already been shown by inhibiting glutamate release and calcium influx, thus preserving the electrochemical gradient and reducing oxidative stress [19]. CoQ10 treatment combined with hypothermia has been shown to improve survival with good neurological outcomes in patients after CA [18]. Edaravone, also known as Radicava, is a member of the substituted 2-pyrazolin-5-one class [20]. Due to its antioxidative and antiapoptotic benefits, Edaravone has been shown to improve survival and neurological outcomes in rats after CA by decreasing brain malonylaldehyde levels and increasing superoxide dismutase activity [21]. N-acetyl cysteine, a thiol-containing compound, functions as an antioxidant and reduces inflammation conferring survival benefits after CA and resuscitation in rats by improving myocardial dysfunction [22]. N-acetyl cysteine has also been shown to protect against neuronal damage by reducing oxidative stress [23]. Vitamin C, also known as ascorbic acid, has antioxidant properties and reduces oxidative stress by scavenging free radicals. High doses of vitamin C have been reported to reduce cerebral and myocardial dysfunction by reducing oxidative stress and inflammatory cytokines after CA in rats [24].

Sodium amobarbital is a short-acting barbiturate and has been shown to reversibly block mitochondrial complex-1 [25]. Blockade of electron transport during ischemic conditions has been shown to protect cardiac mitochondria [25] by reducing mitochondrial superoxide generation and reducing mitochondrial matrix Ca^2+^ overload [26]. Zoniporide has been found effective in sodium-hydrogen exchanger isoform-1 inhibition and helps in reducing cytosolic and mitochondrial Ca^2+^ overload [27]. In an isolated heart model, zoniporide demonstrated cardioprotective effects after ischemia [28]. Furthermore, Poloxamer 188 has been reported as an amphiphilic triblock co-polymer and has the ability to enter injured cell membranes to protect against cell damage after CA [29]. Poloxamer 188 has also been shown to reduce reperfusion injury after myocardial infarction in pigs by reducing cellular and mitochondrial injury [30]. SS-31 interacts with the inner mitochondrial membrane to maintain mitochondrial cardiolipin and maintain mitochondrial function during aging [31]. SS-31 has been reported to increase survival time in rats after CA by potentially reducing mitochondrial damage and maintaining mitochondrial bioenergetics [32,33]. The use of ATP-MgCl_2_ in combination with norepinephrine and vanadate has been reported for cortical protein synthesis after global ischemia in rats [34]. Cyclosporine used during the onset of cardiopulmonary resuscitation has been shown to reduce myocardial dysfunction and improve survival by preventing mitochondrial permeability transition pore (mPTP), thus preventing mitochondrial dysfunction in rabbit models [35]. Cyclosporin treatment also reduces damage after traumatic brain injury in pigs by reducing mPTP and preserving mitochondrial function [36]. Metformin is very well known as an anti-diabetic drug [37]. Studies have demonstrated that administering metformin immediately after CA and resuscitation can enhance survival and provide neuroprotection by decreasing mitochondrial cytochrome c release and mitigating mitochondrial dysfunction [38]. Sulbutiamine is a metabolic supplement [39], and an in vitro study has suggested that sulbutiamine treatment after oxygen-glucose deprivation provides a neuroprotective benefit in rat hippocampal CA1 pyramidal neurons [40].

During the cocktail optimization process, we replaced sodium amobarbital with metformin, which also works as a reversible inhibitor of mitochondrial complex 1 to reduce mitochondrial dysfunction but has a higher safety profile [41]. We removed ATP-MgCl_2_ because of its hypotensive effects despite adding norepinephrine during resuscitation. Thus, the final formulation of our multi-drug cocktail was composed of 10 drugs: cyclosporine A, CoQ10, edaravone, metformin, N-acetyl cysteine, poloxamer 188, SS-31, Sulbutiamine, vitamin C, and zoniporide (see Table 1 for published dosages and approached mechanistic pathways). As CA is a complex disease leading to altered multiple pathways simultaneously producing severe metabolic alteration, these extensive studies related to the therapeutic approach suggest that using these drugs in combination to create a multi-drug cocktail for targeting several altered pathways could be a novel therapeutic approach.

#### 2.1.2. Selection and Formulation of the Multi-Drug Cocktail

We developed a multi-drug cocktail combining pharmacologic agents of varying physical and chemical properties with the objective of obtaining the maximum benefits from the fewest possible number of drugs. In order to produce an effective cocktail, the potential of the included drugs must be assessed to confer either survival or neuroprotection following asphyxial CA in preclinical settings. This information can help in selecting agents that can be effectively and practically combined. We conducted a thorough literature search and selected 10 neuroprotective drugs to comprise our multi-drug cocktail: cyclosporine A, CoQ10, edaravone, metformin, N-acetyl cysteine, poloxamer 188, SS-31, Sulbutiamine, vitamin C, and zoniporide (dosages and targeting pathways are discussed in Table 1). To evaluate any potential adverse effects, such as hypotension or death, each drug was administered to normal rats (*n* = 3) either individually or in various combinations. To arrive at a final formulation, we evaluated sequential iterations of the cocktail with preliminary trials (Appendix A). Except for the 10 drugs, every element of the multi-drug cocktail was included in the vehicle used in our study.

**Table 1 cells-12-01548-t001:** The Ten Pharmaceutical Agents Included in the Multi-drug Cocktail Therapy with their Proposed Mechanism(s) of Action and Dosages.

Group	Agent	Dose, mg/kg	Mechanism(s) of Action	Reference
Antioxidants	CoQ10	30	Increases oxidative stress resistance to, improves mitochondrial bioenergetics	[42]
	Edaravone	3	Reactive oxygen species scavenger: reduces free radical-induced inflammation, reduces oxidative damage and lipid peroxidation to cells minimizes	[21]
	N-acetyl cysteine	150	Antioxidant, anti-inflammatory	[22]
	Vitamin C	50	Protects against oxidative stress, cofactor in catecholamine synthesis	[43]
Calcium and ionic control	Zoniporide	3	NHE-1 inhibitor: protects from ionic gradients, decreases calcium overload from ischemic reperfusion injury	[44]
Lipid membrane regulation	Poloxamer 188	150	Lipid bilayer stabilizer: reduces BBB damage, cerebral edema, and cell death	[45]
	SS-31	0.5	Mitochondrial protectant: protects the mitochondrial membrane by stabilizing cardiolipin	[46]
Mitochondrial protectant and energy production	Cyclosporine A	10	mPTP inhibitor: reduces cytochrome c release, elevate superoxide dismutase activity	[16]
	Metformin	100	Mitochondrial protectant: promotes neurogenesis, protects the BBB, and reduces ROS and inflammation	[38,41]
	Sulbutiamine	12.5	Metabolic supplement for brain energetics: increases thiamine level by easily crossing the BBB	[47]

Abbreviations: ROS indicates reactive oxygen species; BBB = blood–brain barrier; SS-31 = Szeto–Schiller peptide-31; NHE-1 = sodium-hydrogen exchanger isoform; mPTP = mitochondrial permeability transition pore.

### 2.2. Preparation of Cocktail and Vehicle

All hydrophilic drugs (SS-31, metformin, zoniporide, N-acetyl cysteine, vitamin C, edaravone) were dissolved in 750 µL poloxamer 188 (10%). At first, hydrophobic drugs (CoQ10, cyclosporine A, and sulbutiamine) were formulated and then combined with hydrophilic drugs to form the cocktail. To formulate hydrophobic drugs, CoQ10 (15 mg) was mixed with 300 µL polyoxyethanyl-α-tocopheryl sebacate (PTS; 15% *w*/*v* in water; already lyophilized for 24 h). The mixture was then incubated for 15 min at 60 °C using water bath. Following the addition of phosphate-buffered saline (PBS 1X; 1 mL), the mixture was vortexed, incubated for 15 min at 60 °C, and sonicated for 10–15 min. Separately, cyclosporine A (5 mg) was added to 1.5 mL 4% bovine serum albumin (BSA) prepared in 1X PBS, and sonicated for 5 min, while sulbutiamine (6.25 mg) was first dissolved in 100 µL absolute ethanol, then combined with 400 µL polyethylene glycol 300 (PEG-300), and sonicated for 5 min. After completing preparations for each hydrophobic drug separately, the CoQ10 and cyclosporine A formulations were then combined, sonicated for 10 min, combined with the hydrophilic drug solution, and sonicated for another 10 min. Finally, the sulbutiamine formulation was added to the mixture, and the whole solution was sonicated until no solids remained. The formulated cocktail was then filtered through a 0.45 μm syringe filter, and the pH was adjusted to 7.35–7.45 using sodium bicarbonate (50 mEq; 0.8–1.0 mL). The final volume of the multi-drug cocktail was approximately 4.5 mL.

Vehicle was formulated similarly to the multi-drug cocktail; however, no drug was added. PBS (1X; 750 µL) was used as the solubilizing agent. Subsequently, 300 µL of PTS (15% *w*/*v* in water; already lyophilized for 24 h) was incubated at 60 °C for 15 min in a water bath set, followed by addition of 1 mL 1X PBS. The mixture was further incubated at 60 °C for 15 min and then sonicated for 5 min. This mixture was then combined with 1.5 mL BSA (4%; prepared in 1X PBS) and 100 µL absolute ethanol combined with 400 µL PEG-300 and then finally sonicated for 2–3 min. Subsequently, the solution was passed through a 0.45 μm syringe filter to remove impurities, and sodium bicarbonate was employed to adjust the pH to 7.35–7.45.

### 2.3. Evaluation of Active Components in Multi-Drug Cocktail Using Mass Spectrometry

We used mass spectrometry (MS) for our multi-drug cocktail formulation to confirm the stability/molecular interaction of the majority of the drugs. We performed liquid–liquid extraction [48] using ethyl acetate to separate the individual drugs from the multi-drug cocktail formulation. Using electrospray ionization MS (ESI-MS), the molecular masses of the extracted components were analyzed (Figure 1). To get an extract, 500 μL of the multi-drug cocktail was mixed with 1000 μL ethyl acetate, vortexed for 30 s, and centrifuged at 10,000× *g* for 2 min at room temperature. Ethyl acetate was isolated from the top layer and transferred to a new vial. The extraction step was repeated after adding ethyl acetate (1000 µL) to the bottom layer. Then, nitrogen gas was used to dry combined organic phases. The remaining substance was reconstituted in 500 µL methanol, vortexed for 10 s, then centrifuged at 13,000× *g* for 2 min. The resulting extract was then transferred to a polypropylene vial, and a volume of 30 µL from the extract was injected into an LTQ XL™ Linear Ion Trap Mass Spectrometer (Thermo Fisher Scientific, Waltham, MA, USA) for analysis. The spectrometric data were processed with the aid of Xcalibur 3.1 (Thermo Fisher Scientific, Waltham, MA, USA).

### 2.4. Experimental Animal Procedure for Inducing 20 Min of Asphyxial Cardiac Arrest and 30 Min of Cardiopulmonary Bypass Resuscitation

Our animal experiments were conducted following the approved protocol (2016-009) from the Institutional Animal Care and Use Committee (IACUC). To induce asphyxia in male Sprague Dawley rats (400–500 g; Charles River Laboratory, Wilmington, MA, USA), we used procedures that have been previously published [49] (see Appendix A). To minimize the potential outcome variability that might be affected by the gender difference, we only used male rats in this study, as exploring the effect modification caused by the gender difference is beyond the scope of our current study. The rats were divided into two groups: one receiving the multi-drug cocktail treatment and the other receiving vehicle treatment. After inducing 20 min of asphyxial CA, resuscitation was initiated by starting the cardiopulmonary bypass (CPB) flow and resuming ventilation with 100% oxygen. CPB resuscitation continued for 30 min. Immediately after achieving ROSC, multi-drug cocktail/vehicle was injected through femoral vein over 20 min using an infusion pump. We monitored the rats for up to 4 h post-ROSC, and to assess their survival and to evaluate corneal reflex, we gently touched their eyes with a soft cotton gauge and observed by blinking response.

### 2.5. Experimental Animal Procedure for Inducing 12 Min of Asphyxial Cardiac Arrest

All of our experiments were conducted following the approved protocol (2016-009) from the IACUC. We induced asphyxia in male Sprague Dawley rats (400 to 500 g) using the same procedures that have been previously published [49] (see Appendix A). The rats were divided into two groups: one receiving the multi-drug cocktail treatment and the other receiving vehicle treatment. Following surgical preparation and 12 min of asphyxial CA, resuscitation was initiated by chest compressions and resuming mechanical ventilation with 100% oxygen [49]. After ROSC, the rats immediately received either the multi-drug cocktail or vehicle and mechanical ventilation was continued for 2 h post-ROSC. After this, the rats were returned to the animal housing facility and given daily care in accordance with the approved protocol. We monitored the rats for up to 72 h post-ROSC to assess their survival, and a researcher blinded to the study performed neurological evaluation at 24, 48, and 72 h post-ROSC using two scoring scales [50,51]. At 72 h post-ROSC, the rats were deeply anesthetized, euthanized, and their whole brains were removed for histological analysis to observe neuronal degeneration (Figure 2).

### 2.6. Blinded, Randomized, Placebo-Control Study

In order to conduct the experiment in a randomized and blinded manner (Figure 2), Researcher A freshly prepared the multi-drug cocktail and vehicle in a non-identifiable manner. Researcher B randomly assigned the treatment groups, consisting of multi-drug cocktail or vehicle, and was blinded to the experimental conditions. In a blinded manner, Researchers C and D performed the animal surgeries and administered the drugs immediately after successful ROSC. To ensure the quality and consistency of CPR during the resuscitation process, Researcher A performed CPR (via chest compressions) for both treatment groups after 12 min of asphyxial CA. Animals were then transferred to the animal housing facility at 2 h post-ROSC. Neurological evaluation at 24, 48, and 72 h was performed by Researcher E, who was blinded along with the treatment groups.

### 2.7. Neurological Deficit Scores at 24, 48, and 72 h Post-Resuscitation

We used two separate assessment scales (Scale 1 [50] and Scale 2 [51]) to better evaluate neurological outcomes. The modified neurological deficit scale (mNDS) was used to evaluate the general appearance, including consciousness, cranial nerve function, motor skills, sensory abilities, and coordination skills. We made some modifications to the respiration score in Scale 1 to incorporate an intermediate rating of 50, with 100 indicating normal respiration rates between 60 and 120 breaths per minute (BPM), 50 indicating respiration rates between 120 and 140 BPM, and 0 indicating breathing patterns outside of those ranges (Appendix A). The mNDS scores range from 0 to 500, with 0 indicating death and 500 indicating normal status. In addition, we employed a secondary neurological assessment scale (Scale 2), the neurological deficit scale (NDS), to evaluate overall functionality. The NDS scores range from 0 to 80, with 0 representing brain death and 80 representing normal brain function (Appendix A).

### 2.8. Histology and Staining

At 72 h post-ROSC, surviving rats were deeply anesthetized and transcardially perfused with cold phosphate-buffered saline (PBS; 1X, pH = 7.4) to prepare for histology and staining. Whole brains were removed and fixed in 4% paraformaldehyde (PFA) at 4 °C. Afterward, the brains were preserved in 30% sucrose solution in PBS, and serial coronal sections (14 μm) were obtained using a cryostat (CM1900, Leica, USA) and collected on coated glass slides. Staining procedure included Nissl staining (Cresyl Violet, Acros Organics, USA), neuronal nuclear protein (NeuN), and glial fibrillary acidic protein (GFAP) staining (Thermofisher Scientific), which were performed according to published methods [51]. Terminal deoxynucleotidyl transferase dUTP nick end labeling (TUNEL) staining (Abcam, UK) was also performed following the manufacturer’s instructions. Prior to NeuN (neurons) and GFAP (astrocytes) staining, antigen retrieval was performed (Target Retrieval Solution, Dako, pH = 6.0, 1X, Denmark) by incubating the slides in the antigen retrieval solution for 5 min in a pressure cooker. Slides containing brain section were first treated with 0.25% Triton X for 20 min (in 1X TBST), then blocked with 2% normal donkey serum (NDS; VWR, USA) in 1X TBST at room temperature for 1 h. The sections were then incubated with mouse anti-rat NeuN (1:500) and rabbit anti-rat GFAP (1:500) antibodies prepared in 2% NDS in 1X TBST. Slides were incubated overnight at 4 °C with slight agitation. For double immunostaining, slides underwent three washes with 1X TBS for 5 min and were then incubated with secondary antibodies for NeuN (Green, AlexaFluor 488, donkey anti-mouse, 1:400) and for GFAP (Red, AlexaFluor 594, donkey anti-rabbit, 1:400) for 1 h at room temperature. After incubation, the slides were washed three times with 1X TBS followed by water and mounted using Fluoroshield DAPI (Sigma) mounting medium.

The hippocampal CA1 and CA3 regions for sham (no CA/treatment), vehicle, and multi-drug cocktail-treated groups (*n* = 4 each) were imaged bilaterally for Nissl and TUNEL staining in 4 serial coronal sections at 40× magnification, using the BX-X800 bright field microscope (Keyence, Itasca, IL, USA). The number of ischemic (Nissl staining) and apoptotic cells (TUNEL staining) from CA1 and CA3 hippocampal regions were semi-quantified using the BX-X800 Analyzer (Keyence) and averaged from both hemispheres. For NeuN and GFAP, hippocampal CA1 and cortex regions were imaged using the BX-X800 fluorescent microscope (Keyence). Neurons and astrocytes were semi-quantified manually from both CA1 hippocampal and cortex areas using ImageJ, and the average was calculated. All cell counting was conducted per focal area.

### 2.9. Biochemical Analysis

To evaluate plasma cytokines level, blood samples from vehicle- and cocktail-treated rats were withdrawn at baseline and 120 min post-ROSC and centrifuged at 1000× *g* for 10 min to separate the plasma, and the collected plasma was immediately frozen and stored at −80 °C for further biochemical analysis. Plasma was used to measure cytokines (TNFα, IL-6, KC/GRO, and IL-10) levels using a multiplex ELISA as per the manufacturer’s protocol (MSD^®^ MULTI-SPOT Assay System).

### 2.10. Statistical Analyses

The study results are presented as mean ± standard error of the mean (SEM) for continuous variables, while categorical data are reported as proportional frequencies. Hemodynamic parameter measurements (vehicle-treated and cocktail-treated; *n* = 14 each) were compared within and across groups using repeated measures of two-way ANOVA, followed by Tukey’s and Sidak’s corrections for post hoc comparisons, respectively. Lactate and glucose measurements for all rats were analyzed using mixed effects analysis with Tukey’s and Sidak’s corrections for post hoc comparisons. For other analyses, the comparison of two independent groups for continuous variables was made using either an unpaired two-tailed Student’s *t*-test or Mann–Whitney U test as appropriate. The log-rank test was used to compare survival curves of the two groups. Our preliminary analysis of survival (see Appendix A) showed that the mean survival rate in vehicle-treated rats was 25%, and the mean survival rate in cocktail-treated rats was 85% at 72 h following asphyxial CA. For each survival study, we, therefore, anticipated that 14 rats per group would be appropriate to detect a 60% difference in survival between the 2 groups (α = 0.05, β = 0.2 [Power = 80%], two-sided). A statistical significance level of *p* < 0.05 was used. GraphPad Prism version 8.4 (GraphPad Software Inc., La Jolla, CA, USA) and SPSS version 23.0 (SPSS Inc., Chicago, IL, USA) were utilized for statistical analyses.

## 3. Results

### 3.1. Preliminary Analysis of the Multi-Drug Cocktail

Multiple iterations were involved during the development of the multi-drug cocktail formulation. Before generating the final multi-drug cocktail, we conducted eight preliminary trials with different formulations and combinations while keeping the dosages constant (Figure 3 and Appendix A). Each trial provided valuable information regarding the functionality and potential adverse effects of the multi-drug cocktail in both control and CA rats. Through each preliminary trial, areas that needed improvement were identified, such as adjusting the infusion rate of the multi-drug cocktail to minimize hemodynamic fluctuations, adjusting the pH of the multi-drug cocktail with sodium bicarbonate, and removing ATP-magnesium chloride from the formulation due to its hypotensive effect. Despite several failed attempts, the multi-drug cocktail formulation was eventually formulated.

#### 3.1.1. Pilot Study 1: Severe 12 Min of Asphyxial Cardiac Arrest and Cardiopulmonary Resuscitation

Our preliminary results laid the foundation for further assessment of cocktail efficacy post-CA. In an initial unblinded study, which involved vehicle-treated (*n* = 5) and cocktail-treated (*n* = 6) rats, we observed differences in survival and neuroprotective outcomes. After inducing 12 min asphyxial CA, the rats were immediately administered either the vehicle or the cocktail, following which we analyzed differences in the survival, neurological function, hemodynamics, glucose, and lactate between the treatment groups (Appendix A). These preliminary experiments demonstrated a statistically significant improvement in survival at 72 h post-ROSC (*p* = 0.017) and neurological function at 24 h post-ROSC (*p* = 0.015) with multi-drug cocktail treatment.

#### 3.1.2. Pilot Study 2: Highly Lethal 20 Min Model of Asphyxial Cardiac Arrest and Cardiopulmonary Bypass Resuscitation

We further investigated the therapeutic efficacy of our multi-drug cocktail in a highly lethal asphyxial CA model by subjecting rats to 20 min of asphyxial CA and 30 min of CPB resuscitation (*n* = 6 in each group). Rats were kept on mechanical ventilation for 4 h post-ROSC for hemodynamic monitoring and glucose and lactate measurement. The 4 h survival and corneal reflex retention were evaluated. At the culmination of this preliminary experiment, 4/6 cocktail-treated rats surpassed the 4 h survival window, as opposed to only 2/6 vehicle-treated rats. The median survival time was 120.62 min vs. 240.53 min for the vehicle vs. cocktail, respectively (*p* = 0.067), suggesting a trend in increased survival time with cocktail treatment (Appendix A). Furthermore, 4/6 rats demonstrated retention of corneal reflex after 4 h post-cocktail treatment compared with 2/6 rats after vehicle treatment suggesting a trend in increased retention of neurological function with cocktail treatment.

### 3.2. Multi-Drug Cocktail Improves Survival and Mitigates Neurological Dysfunction and Brain Damage after Asphyxial Cardiac Arrest

Rats were subjected to 12 min of asphyxial CA, treated with the cocktail or vehicle upon ROSC, and assessed for survival and neurological function in a fully blinded manner as previously described. The survival rate at 72 h post asphyxial CA was found to be significantly higher in rats treated with the multi-drug cocktail 78.6% (11/14) compared to those treated with the vehicle (28.6%, 5/14) (*p* = 0.006) (Figure 4A), indicating that the multi-drug cocktail therapy significantly improved the survival rate. Surviving rats were evaluated for neurological function at 24, 48, and 72 h post-ROSC with two different neurological scales (mNDS (Scale 1) and NDS (Scale 2); see Appendix A). At 24 h post-ROSC, the mNDS score was significantly higher in surviving rats treated with the cocktail (297.3 ± 17.7) than in those treated with the vehicle (212.5 ± 25.8) (*p* = 0.014) (Figure 4B). At 48 h post-ROSC, although there was a trend in improvement in neurological function between the groups, it was not statistically significant (*p* = 0.082) (Figure 4B). However, at 72 h post-ROSC, the neurological function was significantly higher in rats treated with a multi-drug cocktail (341.4 ± 15.1) compared to those treated with the vehicle (265.0 ± 14.4) (*p* = 0.004) (Figure 4B). With Scale 2, neurological function (NDS score) was only slightly higher in rats treated with the multi-drug cocktail compared to the vehicle-treated rats at 24 h post-ROSC (*p* = 0.190) but significantly improved at 48 (*p* = 0.037) and 72 (*p* = 0.013) h post-ROSC (Figure 4C). These results of the study suggest that the multi-drug cocktail therapy mitigates neurological dysfunction in rats suffering from severe asphyxial CA injury.

The average number of ischemic neurons in the hippocampal CA1 and CA3 regions of the brain was significantly higher in vehicle-treated rats than in sham rats (*p* < 0.0001). However, in cocktail-treated rats, the number of ischemic neurons was significantly lower than in vehicle-treated rats (*p* < 0.05) (Figure 5A). Furthermore, vehicle-treated rats showed a significantly higher number of TUNEL-positive cells in both regions of the hippocampus versus sham rats (*p* < 0.0001), while rats treated with the cocktail had a significantly lower number of TUNEL-positive cells than vehicle-treated rats (*p* < 0.001) (Figure 5B).

Further histological analysis of neurons and astrocytes was performed using NeuN and GFAP staining, respectively. The average number of neurons in the cortex (*p* < 0.01) and CA1 regions (*p* < 0.001) were significantly lower, while the average number of astrocytes in the cortex (*p* < 0.05) was significantly higher in vehicle-treated rats compared to sham (Figure 5C). In cocktail-treated rats, no significant change in neurons or astrocytes was observed in the cortex as compared to sham, while neurons in CA1 regions (*p* < 0.01) were significantly lower compared to sham rats. Although cocktail-treated rats still showed a reduction in the number of neurons as compared to sham, the decrease was not as significant as that observed between sham and vehicle-treated rats (Figure 5C). This supports the survival benefits observed with the cocktail as it may mitigate brain injury up to 72 h after a severe CA injury.

### 3.3. Cocktail Reduces Inflammation and Promotes Anti-Inflammatory Activity after Asphyxial Cardiac Arrest

The effect on global inflammation post-CA as a result of either cocktail or vehicle treatment at 120 min post-ROSC was assessed using rat plasma. Levels of inflammatory markers, such as TNFα, IL-6, KC/GRO, and the anti-inflammatory marker IL-10 did not differ significantly between vehicle-treated and cocktail-treated rats at baseline (Figure 6). However, at 120 min post-ROSC, the plasma concentration of TNFα in vehicle-treated rats (173.20 ± 47.97 pg/mL) increased significantly from baseline (7.85 ± 0.41 pg/mL; *p* = 0.004) and was significantly higher than that of cocktail-treated rats (30.72 ± 11.77 pg/mL; *p* = 0.013) post-ROSC. Similarly, in vehicle-treated rat plasma concentrations of IL-6 (548.30 ± 92.54 pg/mL) and KC/GRO (2615 ± 741.20 pg/mL) significantly increased at 120 min post-ROSC from their respective baseline concentrations (57.57 ± 6.97 pg/mL; *p* < 0.0003 and 54.76 ± 7.09 pg/mL; *p* = 0.005) and were significantly higher than those of cocktail-treated rats (258.2 ± 58.38 pg/mL; *p* = 0.002 and 586.80 ± 79.40 pg/mL; *p* = 0.007) post-ROSC. In contrast, plasma concentrations of TNFα, IL-6, and KC/GRO in cocktail-treated rats experienced a slight and insignificant increase from baseline to 120 min post-ROSC. Furthermore, levels of anti-inflammatory IL-10 in vehicle-treated rats (94.34 ± 15.47 pg/mL) increased significantly at 120 min post-ROSC from its respective baseline (50.34 ± 5.05 pg/mL; *p* = 0.021). However, the increase in IL-10 was significantly higher in cocktail-treated rats from baseline (49.99 ± 2.23 pg/mL) to 120 min post-ROSC (122.0 ± 11.59 pg/mL; *p* = 0.0004).

Cocktail treatment after CA and resuscitation resulted in a significant decrease in the plasma concentration of inflammatory cytokines and an increase in anti-inflammatory cytokines compared to vehicle treatment. Although vehicle-treated rats still showed an increase in the plasma IL-10 level, the increase was much more substantial after cocktail treatment (Figure 6). This suggests that the survival benefits observed with cocktail treatment involve modulating the inflammatory milieu such that the early inflammatory response is dampened while there is potentiation of an anti-inflammatory response after a severe CA injury.

### 3.4. The Cocktail Therapy Resulted in Improved Physiological Characteristics, Hemodynamics, and Arterial Blood Gas Chemistry following Asphyxial Cardiac Arrest

We compared the physiological characteristics of rats treated with the vehicle and cocktail (Table 2 and Table 3). There were no notable variations in their baseline characteristics, time to reach CA, or time to achieve ROSC between the two groups (Table 2). However, at 20 min post-ROSC, cocktail-treated rats had significantly lower partial pressure of oxygen (pO_2_) levels (227.93 ± 30.22 mmHg) compared to vehicle-treated rats (384.08 ± 35.93 mmHg) (*p* < 0.005). At 40 min post-ROSC, oxygen saturation (SaO_2_) was significantly lower in cocktail-treated rats (94.86 ± 1.27%) compared to vehicle-treated rats (98.50 ± 0.36%) (*p* < 0.05) (Table 3). Lactate levels were significantly higher in cocktail-treated rats (4.13 ± 0.15 mmol/l) compared to vehicle-treated rats (3.28 ± 0.25 mmol/l; *p* = 0.032) at 20 min post-ROSC, and this trend was maintained at 40 min post-ROSC (cocktail-treated, 2.87 ± 0.40 mmol/l; vehicle-treated, 1.61 ± 0.21 mmol/l; *p* = 0.042) (Figure 7A). Blood glucose levels were higher in cocktail-treated rats (377.50 ± 21.37 mg/dL) compared to vehicle-treated rats (287.77 ± 18.29 mg/dL) at 20 min post-ROSC (*p* = 0.019), but the levels decreased in both groups until 120 min post-ROSC (Figure 7B). Although there was an initial increase in lactate and glucose levels post-ROSC, both groups showed a decreasing trend toward baseline over time.

At 30 min post-ROSC, mean arterial pressure (MAP) showed an increase in both groups compared to baseline. Vehicle-treated rats showed an increase from 88.41 ± 3.94 mmHg at baseline to 121.64 ± 2.94 mmHg at 30 min (*p* < 0.0001), while cocktail-treated rats increased from 89.07 ± 3.63 mmHg at baseline to 104.38 ± 3.76 mmHg at 30 min (*p* < 0.05) (Figure 7C). However, at 30 min post-ROSC, MAP was significantly lower in cocktail-treated rats (104.38 ± 3.76 mmHg) than in vehicle-treated rats (121.64 ± 2.94 mmHg; *p* < 0.05). By 120 min post-ROSC, MAP returned to baseline values in both groups. Heart rate (HR) was higher in both groups at 120 min post-ROSC compared with baseline values. Vehicle-treated rats showed an increase from 313.79 ± 13.47 beats per minute (bpm) at baseline to 358.54 ± 8.15 bpm at 120 min (*p* < 0.05), while cocktail-treated rats increased from 295.94 ± 12.08 bpm at baseline to 347.74 ± 8.05 bpm at 120 min (*p* < 0.005) (Figure 7D). The HR was only significantly different between the two groups at 60 min post-ROSC (*p* < 0.01). During most of the time course in cocktail-treated rats, the HR remained close to baseline values, while in vehicle-treated rats, HR was highest at 60 min post-ROSC before returning to baseline (Figure 7D).

## 4. Discussion

In this blind randomized control study, we developed a novel multi-drug cocktail therapy and tested its efficacy in a severe-injury model of asphyxial CA. The administration of a multi-drug cocktail immediately after 12 min of asphyxial CA and resuscitation improved 72 h survival with better neurological outcomes. Additionally, the multi-drug cocktail treatment also reduced ischemic and apoptotic neurons in the brain and reduced inflammatory markers in the plasma following 12 min asphyxial CA and resuscitation. We used a severe 12 min asphyxial CA model because it induces a high mortality rate and severe neurofunctional deficits in most survivors. Additionally, this model allows for resuscitation using conventional CPR, which is the mainstay of CA treatment in human patients. Longer durations of CA may require more advanced interventions, such as cardiopulmonary bypass resuscitation, to achieve ROSC. These factors support the use of this model to validate the effectiveness of our multi-drug cocktail therapy. The value of multi-drug therapy has been discussed for a wide range of medical conditions, such as infectious diseases, cancer, diabetes, lipid disorders, and rheumatologic conditions. This approach involves targeting multiple altered metabolic pathways, leading to a more effective outcome response [52,53].

Survival after CA remains poor despite improvements in mechanical, technological, and pharmaceutical interventions [54]. This is due, in part, to the brain’s vulnerability to mitochondrial dysfunction and increased ROS generation [6,10], metabolic disruption [10,11], and neuroinflammation [55] during ischemia–reperfusion injury. Previous studies have evaluated the efficacy of single agents, such as Szeto–Schiller peptide-31 (SS-31) [32], salubrinal [56], or cyclosporine A [16], or combined pharmaceutical agents, such as ATP-MgCl_2_, norepinephrine, and vanadate, or ethanol-epinephrine-vasopressin (HBN-1), or argon-xenon [2,15,34,57] to provide survival and neuroprotection following CA. Consequently, their therapeutic benefit has been limited, and they have not been translated to human patients to date as most have not targeted multiple pathways affected simultaneously after CA. A study in a swine ventricular fibrillation CA model suggested that rapid successive administration of a “cocktail” of epinephrine, vasopressin, amiodarone, and sodium bicarbonate and metoprolol produced more adverse effects and worse short-term outcomes than serial administration of epinephrine, vasopressin, amiodarone, and sodium bicarbonate [58]. Although this study design was not a single formulation but rather a rapid successive administration of the drugs after CA, our blind randomized control study developed a novel multi-drug cocktail therapy and tested its efficacy in a severe asphyxial CA model, with a focus on neuroprotection. The cocktail was designed to target multiple altered pathways simultaneously and needs further exploration. This study supports the shift towards multi-target cocktail therapy to improve outcomes after CA.

The process of formulating our multi-drug cocktail involved combining various drugs that have individually shown some degree of neuroprotection against ischemia–reperfusion injury [16,32,59,60,61]. Developing a single cocktail from several different drugs can be challenging as it requires finding the right balance between therapeutic efficacy and minimizing potential adverse effects. Throughout each stage of cocktail formulation, we carefully assessed this balance. Once the final multi-drug cocktail formulation was developed, we used MS analysis to verify that the majority of the drugs, including cyclosporine A, edaravone, metformin, N-acetyl cysteine, vitamin C, and Zoniporide, were preserved in the formulation. Our MS analysis did not detect any additional peaks, apart from traces of the formulation solvent PEG [62], suggesting that no detectable drug–drug interactions exist in the formulation.

In addition to survival and neurological outcomes, we monitored changes in hemodynamics and blood chemistry following the administration of the multi-drug cocktail after asphyxial CA. During the development of the cocktail, several formulations caused adverse effects, which led to poor outcomes. However, through various trials, we developed a multi-drug cocktail formulation that maintained hemodynamics and blood chemistry parameters close to baseline levels. This is crucial because CA causes severe physiological abnormalities, and survival depends on reducing their severity. The possible explanation could be that the cocktail may help the body to regain metabolism and normal physiological functioning rather than having excess oxygen, which may be reflected in the slightly lower oxygen saturation after cocktail administration. Furthermore, our cocktail has many mitochondrial protectants that could allow for more normal mitochondrial function but needs further exploration. Although the multi-drug cocktail achieved this goal, there was a significant increase in lactate levels, suggesting the need for further improvement.

After asphyxial CA and resuscitation, multi-drug cocktail treatment significantly improved 72 h survival and neurological function and ameliorated brain morphology via preservation of brain cell morphology, reduced apoptosis, neuronal retention, and astrocyte number reduction (Figure 5). CA results in neuronal cell death via apoptosis, necrosis, and other cell death pathways, and preventing loss of brain function post-ROSC is dependent upon minimizing the degree of cellular loss [38,56,63]. Furthermore, inflammatory processes start immediately after ischemia and lead to cerebral injury [64,65] through several phases. It includes an influx of peripheral immune cells, activation of glial cells, and the release of pro-inflammatory mediators [66,67,68]. These inflammatory processes cause neuronal damage, exacerbate vasomotor dysregulation, endothelial dysfunction, and disrupted blood–brain barrier (BBB) [69]. Studies have suggested that reducing inflammatory processes provides neuroprotection after the ischemic injury [70]. TNFα exposure has also been shown to cause neuronal mitochondrial dysfunction [71]. In this study, we observed that cocktail treatment after CA significantly reduced the pro-inflammatory cytokines—TNFα, IL-6, and KC/GRO—compared with vehicle treatment. Furthermore, an increase in the anti-inflammatory cytokine IL-10, compared with vehicle treatment, was also observed in the cocktail treatment group. IL-10 released by T helper 2 (Th2) lymphocytes can reduce the effect of TNFα [72]. In animal models and human patients, decreased levels of IL-10 have been shown to develop large infarcts after focal ischemia [73,74]. As such, our results suggest that the cocktail provides neuroprotection following CA by both reducing inflammatory mediators and upregulating anti-inflammatory mediators.

The process of discovering a new drug is a long and resource-intensive endeavor that can take up to 12 years [75]. Therefore, we propose a more feasible approach of repurposing well-studied agents [4,21,28,29,32,35,38,40,42] into a single cocktail formulation. Our study builds on prior attempts at post-CA neuroprotection and shows the potential of a multi-drug cocktail approach for improving outcomes after asphyxial CA. However, our study has several limitations. We did not compare outcomes using the individual drugs because the goal of the study was to target multiple altered pathways simultaneously. Additionally, the mechanisms of action for each constituent in the cocktail need further exploration. While it is expected that the drugs will function as intended to confer therapeutic benefit, there may be additional mechanisms of action as well that are either unknown or require further investigation when administered in this form. Third, our MS analysis could not detect CoQ10, SS-31, sulbutiamine, and poloxamer 188 because these drugs are structurally complex and highly heterogeneous. We are further working on improving our methodology to identify and confirm the stability of all the drugs in the cocktail. Finally, since survival post-CA also depends on the recovery of other organs, we aim to investigate whether our cocktail could ameliorate and protect against ongoing damage in other tissues.

Despite these limitations, this is the first study for the development of a multi-drug cocktail therapy that combines 10 distinct pharmaceutical agents and rigorously tests its efficacy in a rodent model of severe asphyxial CA. Although the exact mechanisms underlying our cocktail have yet to be elucidated, our results support simultaneous treatment of multiple affected pathways via cocktail therapy for neuroprotection and survival post-CA.

## 5. Conclusions

Multi-drug cocktail therapy could be a better therapeutic approach to combat multiple altered pathways following ischemia–reperfusion injury caused by asphyxial cardiac arrest.

## Figures and Tables

**Figure 1 cells-12-01548-f001:**
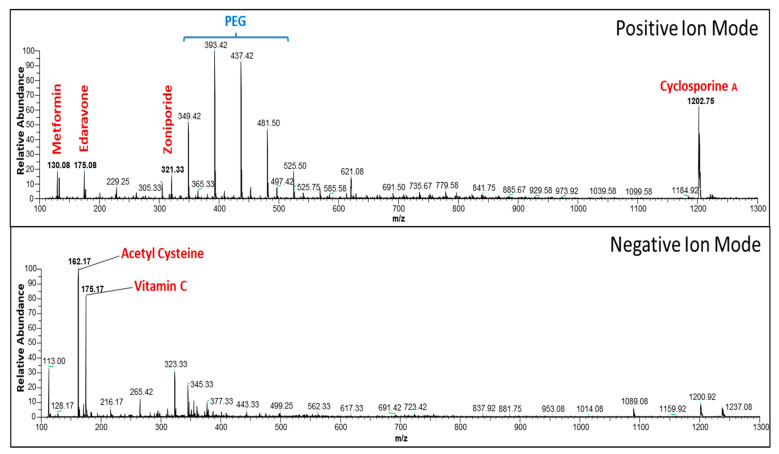
The 10-drug combination that was used to develop a multi-drug cocktail was analyzed using electrospray ionization/mass spectrometry (ESI-MS). The analysis was carried out in both positive and negative ion modes following an ethyl acetate extraction of the multi-drug cocktail. The results of the analysis provided information on the mass/charge (m/z) values of the primary constituents of the cocktail, including cyclosporine A, edaravone, metformin, N-acetyl cysteine, vitamin C, and zoniporide. Additionally, traces of polyethylene glycol (PEG) were also detected during the analysis.

**Figure 2 cells-12-01548-f002:**
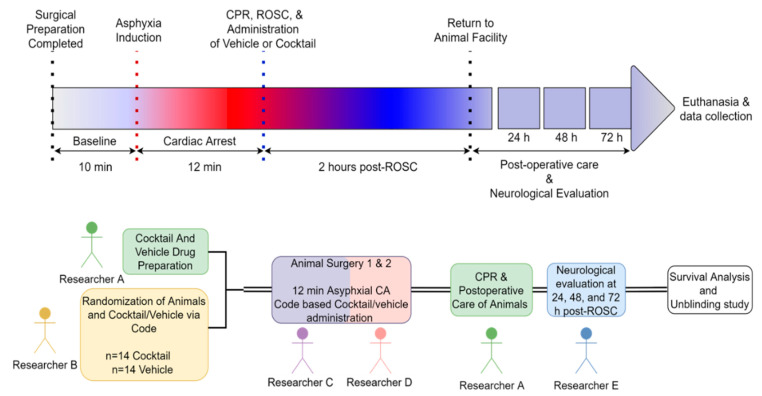
Schematic diagram outlines the sequential steps and timeframe for carrying out blinded experiments. The top schematic section displays the chronological order of the rat asphyxial CA experiment, which extends up to 72 h survival. The bottom section shows the procedure used to conduct the survival study in a blinded manner. CPR = cardiopulmonary resuscitation; ROSC = return of spontaneous circulation.

**Figure 3 cells-12-01548-f003:**
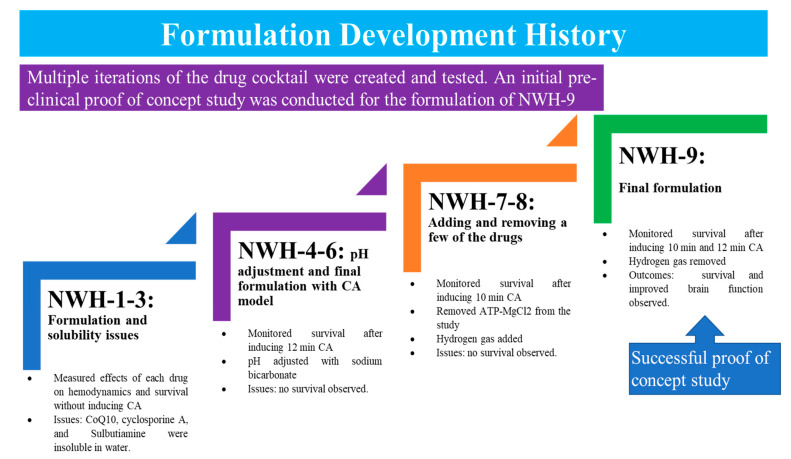
Testing process for cocktail iterations. The figure illustrates the multiple iterations (NWH-1 through NWH-9) that were prepared before reaching the final formulation. NWH = Northwell Health; CA = Cardiac Arrest; PBS = Phosphate-buffered saline.

**Figure 4 cells-12-01548-f004:**
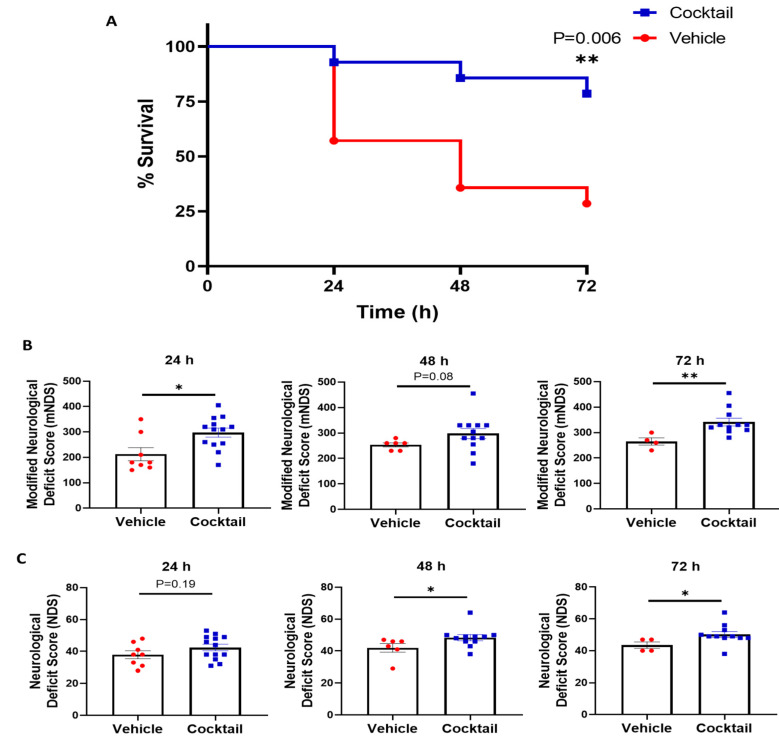
The survival curve and neurological deficit score suggesting multi-drug cocktail significantly improves survival and neurological function following asphyxial cardiac arrest and resuscitation. (**A**) At 72 h after ROSC, the survival rate was 78.6% in multi-drug cocktail-treated rats compared to 28.6% in vehicle-treated rats (*p* = 0.006). (**B**) The mNDS (Scale 1) scores at 24, 48, and 72 h post-ROSC show there was a significant improvement in multi-drug cocktail-treated rats compared to vehicle-treated rats at 24 and 72 h post-ROSC, with a trend towards improvement at 48 h. (**C**) With Scale 2, the neurological function scores were slightly higher in multi-drug cocktail-treated rats at 24 h post-ROSC and significantly improved at 48 h and 72 h post-ROSC compared to vehicle-treated rats. Dead rats (score = 0) were excluded from the statistical analysis. Data are shown as mean ± SEM. * *p* < 0.05, ** *p* < 0.005. ROSC = return of spontaneous circulation; mNDS = modified neurological deficit score; NDS = neurological deficit score.

**Figure 5 cells-12-01548-f005:**
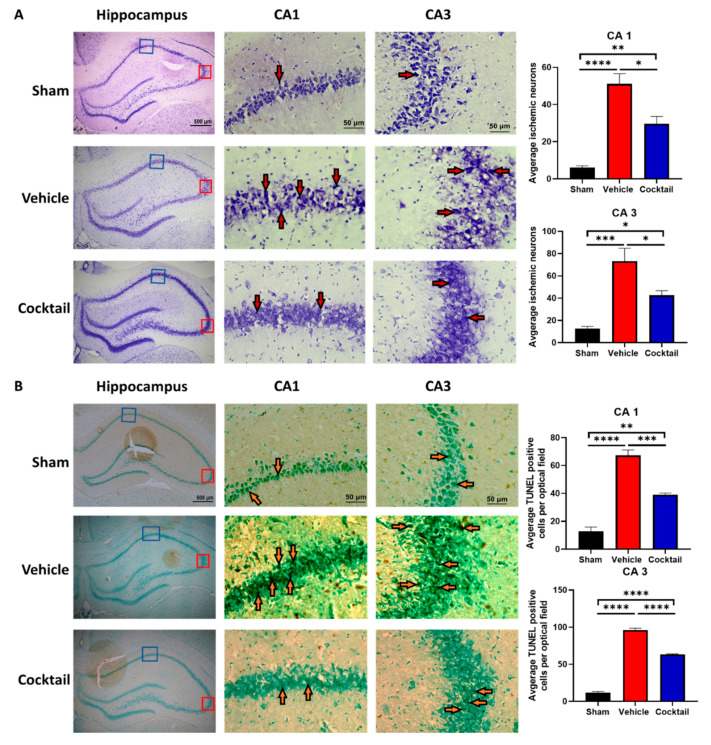
Histological staining of brain tissue showing cocktail treatment mitigates brain injury after asphyxial cardiac arrest and resuscitation. Nissl (**A**) and TUNEL (**B**) staining indicates the presence of ischemic cells (red arrow) and apoptotic cells (orange arrow), respectively, in CA1 and CA3 regions of hippocampus. The number of ischemic neurons and apoptotic cells in the CA1 and CA3 regions were significantly higher in vehicle-treated rats compared to sham rats. However, treatment with the cocktail significantly reduced the number of ischemic neurons and apoptotic cells in these regions when compared to vehicle-treated rats. (**C**) NeuN-stained neurons (green), GFAP-stained astrocytes (red), and DAPI-stained nonspecific cellular nuclei (blue) in the cortex and hippocampal CA1 regions. A significant loss of neurons in neurons was observed in both the cortex and CA1 of vehicle-treated rats compared to sham, while cocktail treatment demonstrated varying degrees of neuronal preservation in both brain regions. The number of astrocytes significantly increased in the cortex when comparing sham to vehicle-treated rats, while no significant differences were observed between cocktail-treated and sham rats. Data are shown as mean ± SEM. * *p* < 0.05, ** *p* < 0.01, *** *p* < 0.001, and **** *p* < 0.0001.

**Figure 6 cells-12-01548-f006:**
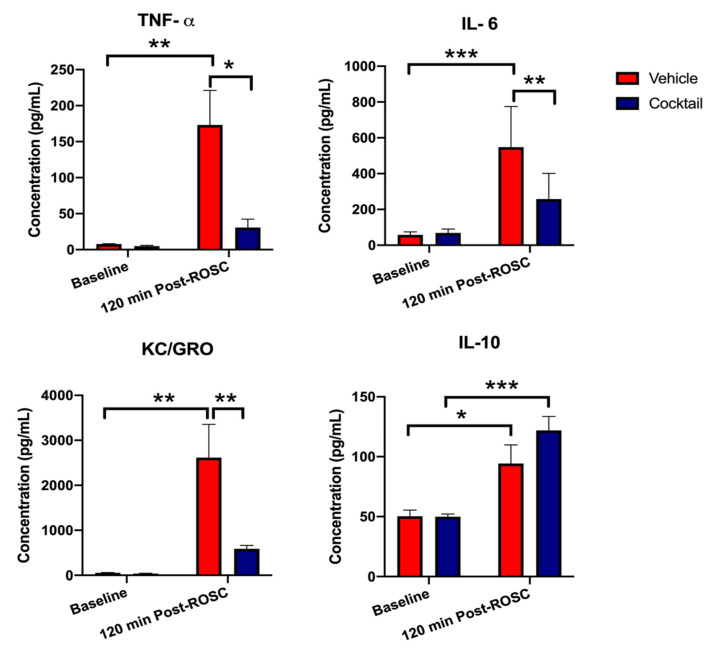
Changes in inflammatory and anti-inflammatory markers after asphyxial cardiac arrest and resuscitation. Cocktail treatment reduces inflammation and improves anti-inflammatory activity after asphyxial cardiac arrest and resuscitation. Plasma TNFα, IL-6, KC/GRO, and IL-10 levels significantly differed between vehicle-treated and cocktail-treated rats at 120 min after return of spontaneous circulation (ROSC). Plasma concentration of TNFα, IL-6, and KC/GRO significantly increased from baseline after cardiac arrest and resuscitation in vehicle-treated rats, while these inflammatory cytokines were significantly reduced in cocktail-treated rats. Plasma concentration of anti-inflammatory IL-10 significantly increased after cardiac arrest in vehicle-treated rats; however, the increase was more prominent in cocktail-treated rats. Data are shown as mean ± SEM. * *p* < 0.05, ** *p* < 0.005, *** *p* < 0.0005.

**Figure 7 cells-12-01548-f007:**
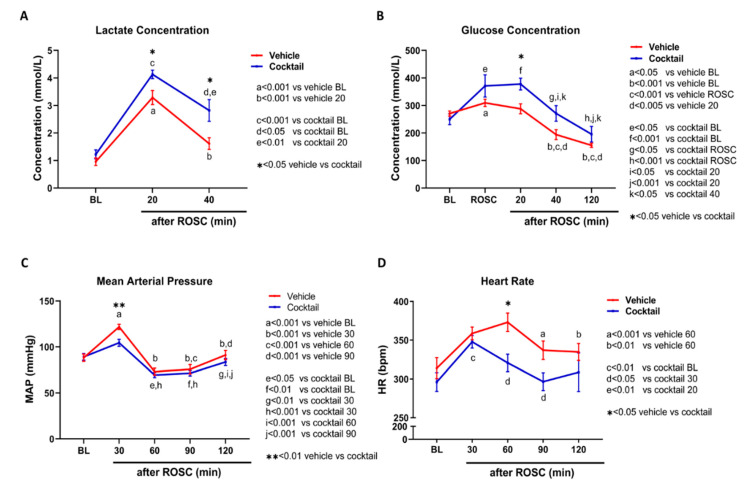
The effect of cocktail or vehicle treatment on blood lactate, glucose, and hemodynamics following asphyxial cardiac arrest and resuscitation in rats. (**A**) Both vehicle-treated and cocktail-treated rats had significantly higher lactate concentrations at 20 and 40 min post-ROSC compared to their respective baselines. Lactate concentration was also found to be greater in cocktail-treated rats than in vehicle-treated rats at 20 and 40 min post-ROSC. (**B**) Glucose concentration was higher in cocktail-treated rats than in vehicle-treated rats, with the most significant difference observed at 20 min post-ROSC. However, glucose concentration decreased over time in both groups. (**C**) MAP was significantly greater in vehicle-treated rats than in cocktail-treated rats only at 30 min post-ROSC. (**D**) HR in vehicle-treated rats was significantly higher at 60 min post-ROSC compared to cocktail-treated rats (**D**). These findings are presented as mean ± SEM, and significant comparisons with each group are shown in the inserts. * *p* < 0.05, ** *p* < 0.01 between vehicle and cocktail. ROSC = return of spontaneous circulation; BL = baseline; MAP = mean arterial pressure; BPM = beats per minute.

**Table 2 cells-12-01548-t002:** Baseline Characteristics of Arterial Blood Gas analysis in Rats Treated with Either Vehicle or Cocktail after Asphyxial Cardiac Arrest ^†^.

	Vehicle (*n* = 14)	Cocktail (*n* = 14)	*p* Value
**Baseline characteristics**			
Body weight (g)	450.6 ± 6.9	458.1 ± 7.2	0.46
Mean arterial pressure (mmHg)	88.1 ± 4.0	89.1 ± 3.6	0.86
Heart rate (bpm)	313.8 ± 13.5	295.9 ± 12.1	0.33
Esophageal temperature (°C)	36.8 ± 0.1	36.9 ± 0.1	0.59
**Cardiac arrest characteristics**			
Time to cardiac arrest (sec)	184.0 ± 6.6	193.4 ± 6.6	0.27
CPR time to ROSC (sec)	57.4 ± 2.2	57.9 ± 2.6	0.93

Abbreviations: CPR = cardiopulmonary resuscitation; bpm = beats per minute; ROSC = return of spontaneous circulation. ^†^ Data expressed as mean ± SEM.

**Table 3 cells-12-01548-t003:** Analysis of Arterial Blood Gas in Rats Treated with Vehicle or Cocktail after Asphyxial Cardiac Arrest ^†^ at Baseline, 20 Min Post-ROSC, and 40 Min Post-ROSC.

		Time after ROSC	
	Baseline	20 Min Post-ROSC	40 Min Post-ROSC
**pH**			
Vehicle	7.42 ± 0.01	7.23 ± 0.01	7.34 ± 0.01
Cocktail	7.40 ± 0.01	7.18 ± 0.02	7.31 ± 0.02
**pCO_2_ (mmHg)**			
Vehicle	39.91 ± 1.60	52.23 ± 2.59	41.25 ± 1.64
Cocktail	43.36 ± 1.16	54.84 ± 3.28	41.80 ± 2.01
**pO_2_ (mmHg)**			
Vehicle	113.42 ± 7.97	384.08 ± 35.93	128.08 ± 9.33
Cocktail	109.71 ± 6.79	227.93 ± 30.22 **	106.29 ± 17.44
**HCO_3_^−^ (mEq/L)**			
Vehicle	25.86 ± 0.85	21.99 ± 0.56	22.13 ± 0.43
Cocktail	27.21 ± 0.53	20.62 ± 0.63	21.20 ± 0.47
**SaO_2_ (%)**			
Vehicle	98.08 ± 0.42	99.33 ± 0.67	98.50 ± 0.36
Cocktail	98.00 ± 0.38	97.36 ± 1.42	94.86 ± 1.27 *

Abbreviations: ROSC = return of spontaneous circulation; CPR = cardiopulmonary resuscitation; pCO_2_ = partial pressure of carbon dioxide; pO_2_ = partial pressure of oxygen; HCO_3_^−^ = bicarbonate; SaO_2_ = oxygen saturation. * *p* < 0.05 and ** *p* < 0.005 vs. vehicle. ^†^ Data expressed as mean ± SEM.

## Data Availability

On reasonable request, the data that support the findings of this study are available from the corresponding author.

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
