# Peer review of "Multi-Drug Cocktail Therapy Improves Survival and Neurological Function after Asphyxial Cardiac Arrest in Rodents"

_cells, 2023, doi:10.3390/cells12111548_

Round 1

Reviewer 1 Report

In this study, Choudhary et al developed a therapeutic cocktail drug capable of targeting multiple pathways of ischemia-reperfusion injury after cardiac arrest and they showed its effectiveness in improving neurologically-intact survival in rats.

The study is very interesting, well described, and figures are very appealing. There are only few comments to address.

Major comments

1)      Paragraph 2.10 should be reduced in length.

2)      Caption of Figure 4 should be more descriptive. (e.g., survavial curves….instead of saying “cocktail improves…”). The same for Figure 5, 6

3)      You should start the discussion with your findings and after explaining the basis and the comparisons.

4)      You should include a Conclusion paragraph, briefly summarizing your message.

5)      Number of funding should be inserted as well as number of protocol and approval by committee.

Minor comments

1)      spaces and punctuation to be corrected throughout the text

2)      spelling errors (e.g., “survival” instead of “surivival”)

3)      Correspondence to is repeated twice on the first page.

minor spelling errors

Author Response

Major comments

1)      Paragraph 2.10 should be reduced in length.

Response: We appreciate the reviewer’s suggestion, we have updated the paragraph, in fact, we have updated all the sections.  

2)      Caption of Figure 4 should be more descriptive. (e.g., survival curves….instead of saying “cocktail improves…”). The same for Figure 5, 6

Response: We appreciate the reviewer’s suggestion, we have updated the caption for Figure 4, in fact, we have updated all the sections. 

3)      You should start the discussion with your findings and after explaining the basis and the comparisons.

Response: We appreciate the reviewer’s suggestion; we have updated the discussion section.

4)      You should include a Conclusion paragraph, briefly summarizing your message.

Response: We appreciate the reviewer’s suggestion; we have added the conclusion section summarizing our message.

5)      Number of funding should be inserted as well as number of protocol and approval by committee.

 Response: We appreciate the reviewer’s suggestion; we have added the protocol number. Since the project was funded by Industry support, we haven’t provided the grant number.

Minor comments

1)      spaces and punctuation to be corrected throughout the text

Response: We are sorry for the spelling errors; we updated the manuscript.

2)      spelling errors (e.g., “survival” instead of “surivival”)

 Response: We are sorry for the spelling errors; we updated the manuscript

3)      Correspondence to is repeated twice on the first page.

 Response: We are sorry for the mistake and appreciate the reviewer’s comment; we have deleted the repeated “correspondence to” word.

Reviewer 2 Report

This is an interesting pre-clinical study examining intracerebral ischemia-reperfusion injury in a rodent model of cardiac arrest. The authors utilized a multi-drug cocktail for neuroprotection. The individual therapeutic agents used in this cocktail were previously reported to be beneficial. Importantly, the authors performed Mass Spect analysis to ascertain the active cocktail components. The study protocol which is Blinded, Randomized, and Placebo-Controlled, and the procedures and timelines are well developed and remain the key strength of this study. I have some comments-

1. Though described as a cardiac arrest model, a careful review of the study protocol suggests this to be more of an asphyxia model. It is unclear whether there was VT/VFib cardiac arrest. If not, the mechanisms of intracranial injury should be explained differently.  Global asphyxia is associated with a different pathology compared to an arrhythmic cardiac arrest. The reversal of such pathologies is also done differently. This needs to be clearly explained in the methods.

2. Accordingly, what parameters guided the 12 min CPR and ROSC?  

3. Although the goal of this study was to examine the combined effect of a cocktail regimen, it can be challenging to translate this to clinical practice. Multiple agents are involved and their individual effects and drug interactions can be a challenge, especially in subjects with other co-morbidities.  It would be more helpful to provide a more simplified explanation of the "formulation development history".

4. It is unclear whether the cocktail mitigates the primary ischemic injury or attenuates the reperfusion injury. If it works at both levels, a possible mechanistic explanation should be provided.

5. In the discussion, the authors state that the cocktail is helping the body to regain metabolism and cocktail has many mitochondrial protectants. These statements are very broad and not fully supported by the data presented in this manuscript.

Author Response

Comments and Suggestions for Authors (Reviewer 2 )

This is an interesting pre-clinical study examining intracerebral ischemia-reperfusion injury in a rodent model of cardiac arrest. The authors utilized a multi-drug cocktail for neuroprotection. The individual therapeutic agents used in this cocktail were previously reported to be beneficial. Importantly, the authors performed Mass Spect analysis to ascertain the active cocktail components. The study protocol which is Blinded, Randomized, and Placebo-Controlled, and the procedures and timelines are well developed and remain the key strength of this study. I have some comments-

  1. Though described as a cardiac arrest model, a careful review of the study protocol suggests this to be more of an asphyxia model. It is unclear whether there was VT/VFib cardiac arrest. If not, the mechanisms of intracranial injury should be explained differently.  Global asphyxia is associated with a different pathology compared to an arrhythmic cardiac arrest. The reversal of such pathologies is also done differently. This needs to be clearly explained in the methods.

Response: We appreciate the reviewer’s suggestion; we have rewritten the study protocol indicating the study to be an asphyxial cardiac arrest. 

Although the reviewer's comments regarding the initial reversal of VT/VF vs asphyxia being different are correct, this difference in intervention only exists in the early phase when the ventricular arrhythmia occurs; however, most patients who do suffer VT/VF arrest eventually have an asystolic cardiac rhythm at which point physicians are still continuing the resuscitative effects. During this time, the patient is also suffering a massive, global hypoxic injury. The asphyxial arrest also has a phase in which the cardiac rhythm has a VT/VF-like rhythm and then eventual asystole. Essentially both types of etiologies lead to massive brain injury, and both can occur in humans. However, inducing VT/VF in animal models leads to substantial heart damage due to the electric current utilized, and therefore, our lab uses the more consistent model of asphyxia. It is a very well-accepted model and therefore we feel that VT/VF model does not need to be addressed. However, we further clarify details about the asphyxia model in the manuscript.

  1. Accordingly, what parameters guided the 12 min CPR and ROSC? 

Response: We appreciate the reviewer’s concern. The methods section details the parameters of this model. This is a well-developed model used by multiple laboratories around the world. After we initiate asphyxia, CA is defined as when the cardiovascular system is unable to maintain a BP >20 mmHg. This is insufficient pressure to perfuse organs. ROSC is defined as BP > 60 mmHg after the initiation of CPR 12 mins post-initiation of asphyxia.

Furthermore, we have also discussed about choosing 12 min CA, CPR, and ROSC in the discussion section “ We used a severe 12 min asphyxial CA model because it induces a high mortality rate and severe neurofunctional deficits in most survivors. Additionally, this model allows for resuscitation using conventional CPR, which is the mainstay of CA treatment in human patients”.

  1. Although the goal of this study was to examine the combined effect of a cocktail regimen, it can be challenging to translate this to clinical practice. Multiple agents are involved and their individual effects and drug interactions can be a challenge, especially in subjects with other co-morbidities.  It would be more helpful to provide a more simplified explanation of the "formulation development history".

Response: We appreciate the reviewer’s concern.  The formulation development was an arduous process in which multiple combinations of the drugs were made and actively tested in animals for adverse effects. We agree with the reviewer that the multiple agents are complex. However, as we highlighted in the paper, these drugs either are being used in patients and/or have been tested for neuroprotection individually. Additionally, our Mass spec result did not detect any additional peak suggesting no further cross-reactivity after combining these multiple drugs. Therefore, we had higher confidence in their additive effects by combining them. However, we did our due diligence by doing various trials to ensure that we are mitigating adverse effects (i.e: we removed ATP-MgCl from the cocktail due to its negative effects). Finally, the whole concept of this study is that smaller, more simplified drug therapies have not been beneficial to patients as targeting multiple altered pathways certainly requires multiple combinations of drugs which can work synergistically. So, we should try a more thorough compilation as we did in this study.

  1. It is unclear whether the cocktail mitigates the primary ischemic injury or attenuates the reperfusion injury. If it works at both levels, a possible mechanistic explanation should be provided.

Response: We appreciate the reviewer’s concern. The drugs we used in this study are either being used or have already shown benefits in animal models. By combining these drugs, our goal was that the drugs would be enacting their intended effects in order to protect against ischemia-reperfusion injury. To clarify for the reviewer, reperfusion injury cannot exist without having an ischemic injury beforehand. As such, reperfusion injury supplements injury sustained after ischemia. Our cocktail was designed with this in mind as most researchers in CA are aware of the relationship between ischemia and reperfusion in their combined and interrelated effects. It is possible that our drugs may have other added effects due to the various other drugs in the cocktail which is a goal of future studies. This has all been described in the discussion section.

  1. In the discussion, the authors state that the cocktail is helping the body to regain metabolism and cocktail has many mitochondrial protectants. These statements are very broad and not fully supported by the data presented in this manuscript.

Response: We appreciate the reviewer’s concern. If we look at the survival/neuroprotective benefits, improved hemodynamics, and the fact that many drugs in the cocktail are known mitochondrial protectants, this statement is not too far from the observed data. However, we have made this statement less broad as the reviewer suggests.

Round 2

Reviewer 2 Report

My concerns have been addressed. Thank you. 

Author Response

We appreciate the reviewer's response and suggestions.